# Systems Biology Approach for Personalized Hemostasis Correction

**DOI:** 10.3390/jpm12111903

**Published:** 2022-11-15

**Authors:** Alexandra S. Pisaryuk, Nikita M. Povalyaev, Alexander V. Poletaev, Alexey M. Shibeko

**Affiliations:** 1City Clinical Hospital named after V.V. Vinogradov, 117292 Moscow, Russia; 2Medical Institute, Department of Internal Medicine, Peoples’ Friendship University of Russia (RUDN), 117198 Moscow, Russia; 3Dmitry Rogachev National Medical Research Center of Pediatric Hematology, Oncology and Immunology, 117198 Moscow, Russia; 4Laboratory of Cell Physiology and Biophysics, Center for Theoretical Problems of Physicochemical Pharmacology, 30 Srednyaya Kalitnikovskaya Street, 109029 Moscow, Russia

**Keywords:** blood coagulation, mechanism-driven models, PK/PD modeling, personalized medicine

## Abstract

The correction of blood coagulation impairments of a bleeding or thrombotic nature employs standard protocols where the type of drug, its dose and the administration regime are stated. However, for a group of patients, such an approach may be ineffective, and personalized therapy adjustment is needed. Laboratory hemostasis tests are used to control the efficacy of therapy, which is expensive and time-consuming. Computer simulations may become an inexpensive and fast alternative to real blood tests. In this work, we propose a procedure to numerically define the individual hemostasis profile of a patient and estimate the anticoagulant efficacy of low-molecular-weight heparin (LMWH) based on the computer simulation of global hemostasis assays. We enrolled a group of 12 patients receiving LMWH therapy and performed routine coagulation assays (activated partial thromboplastin time and prothrombin time) and global hemostasis assays (thrombodynamics and thrombodynamics-4d) and measured anti-Xa activity, fibrinogen, prothrombin and antithrombin levels, creatinine clearance, lipid profiles and clinical blood counts. Blood samples were acquired 3, 6 and 12 h after LMWH administration. We developed a personalized pharmacokinetic model of LMWH and coupled it with the mechanism-driven blood coagulation model, which described the spatial dynamics of fibrin and thrombin propagation. We found that LMWH clearance was significantly lower in the group with high total cholesterol levels. We generated an individual patient’s hemostasis profile based on the results of routine coagulation assays. We propose a method to simulate the results of global hemostasis assays in the case of an individual response to LMWH therapy, which can potentially help with hemostasis corrections based on the output of global tests.

## 1. Introduction

The basis of blood coagulation is a cascade of reactions in which coagulation factors participate [1]. The physiological function of blood clotting is to stop bleeding (hemostasis). It is initiated by tissue factor (TF), a membrane glycoprotein exposed on the subendothelium, which is exposed when the vessel is damaged. Coagulation factors form active complexes on the negatively charged phospholipid surface, providing an increase in the enzymatic activity of free factors by 2–3 orders of magnitude. In the body, the source of such membranes seems to be microvesicles, activated platelets and plasma lipoproteins [2]. The resulting reaction of plasma blood coagulation is the thrombin-induced activation of fibrinogen, which polymerizes to form a fibrin clot [3]. 

The impairment of blood coagulation can manifest itself not only as a standalone disease, such as deep vein thrombosis. Often, it accompanies oncological [4], immunological [5] and infectious [6] diseases; it can reveal itself during surgery [7,8] or hormonal therapy [9]. The most common way to correct hemostasis is to use standard protocols, in which an anticoagulant drug (in the case of thrombotic complications) is administered at a dose corresponding to the patient’s weight. However, this approach does not always lead to hemostasis correction, as it does not take into consideration the individual characteristics of the patient, such as the sensitivity to the drug, different initial states of hemostasis and differences in pharmacokinetics. Thus, in the case of individual therapy adjustment, its efficacy and drug dosing are monitored by means of hemostasis tests, both routine (activated partial thromboplastin time (*APTT*) and international normalized ratio (INR) [10]) and global (thrombin generation assay, thromboelastogram and thrombodynamics [11]). Global assays allow for the resolution of more details of the picture of the hemostasis state and not only show a general shift towards bleeding or thrombosis but also find the impaired part of the hemostatic system [12]. Monitoring hemostasis with such tests during therapy adjustment allows physicians to figure out whether the current drug is suitable for the current patient and what dosage is needed to normalize hemostasis. Such an approach sufficiently increases the treatment quality, but it also increases its cost and requires a specialist who can properly interpret the results of the test and who will be able to make the correct therapy adjustment based on these results.

One of the possible cost-effective approaches towards personalized medicine is to use computational models of blood coagulation. Such an approach was realized in the works of K.E. Brummel-Ziedins’ group, where an in silico model of thrombin generation [13] was used to predict in vitro thrombin generation in different groups: healthy volunteers [14], patients with deep vein thrombosis [15], and patients with acute ischemic stroke or transient ischemic attack [16]. Another group using the same computational model of thrombin generation investigated the effect of prothrombin complex concentrate on the correction of thrombin generation in dilution-induced coagulopathy [17] and the effects of a reduced temperature on thrombin generation [18].

The advantages of such an approach are a reduction in the cost of therapy, a reduction in blood collection and the minimization of the stage of drug selection and dose adjustment.

Heparins are widely used in many pathological states, including cardiovascular diseases, surgery and the post-operative prophylaxis of thromboembolic complications. Low-molecular-weight heparins (LMWHs) have stable pharmacokinetics, with almost 100% bioavailability. Many specialists believe that constant coagulation monitoring during therapy is not required because clinical doses of LMWHs may be corrected based solely on the patient’s body weight [19]. However, the heparin concentration in blood also varied substantially in a healthy population and was only partially correlated with the donor’s body weight [20]. A more precise dose correction may be required in many scenarios (e.g., low or high body mass index; critical states; renal failure (creatinine clearance < 30 mL/min); when switching from one anticoagulant to another; and age (children and older adults > 75 years) [19,21,22]. The frequency of bleeding and thrombotic complications during LMWH therapy is 3–5% [23].

In this work, we used a model of blood coagulation that described more than a hundred reactions to predict the results of the global hemostasis test for patients with different coagulation disorders undergoing LMWH therapy.

In order to develop an individual coagulation profile of the patient, we estimated the concentration of the main coagulation factors and inhibitors based on the results of routine hemostasis assays, such as *APTT* and *PT*. The individual pharmacokinetic profile of LMWH was based on the developed model described herein, which considers its clearance and distribution based on creatinine clearance and the cholesterol level. After this profile was developed, a comparison of simulations of hemostasis correction and experimental global hemostasis assays was carried out, and we found a good correlation between our simulation-based predictions and the experimentally measured hemostasis state of a patient.

## 2. Methods

### 2.1. Study Population

We conducted an observational study in 12 patients with IE admitted to the City Clinical Hospital named after V.V. Vinogradov, Moscow, Russia. This study enrolled patients older than 18 years old undergoing LMWH therapy. Enoxaparin sodium 4000 IU (and in 1 case, fondaparinux sodium 2.5 mg) was administered subcutaneously once or twice a day.

Blood samples were collected at three time points: 3 h after the injection of LMWH, 6 h after the injection of LMWH, and before the next injection of LMWH (12 h after the injection of LMWH).

The blood of healthy volunteers (N = 5) was collected for the preparation of normal pooled plasma (NPP).

Blood was drawn into vacuum tubes (Monovette, Sarstedt, Nümbrecht, Germany) with 106 mM sodium citrate buffer (pH 5.5) at a 9:1 blood:anticoagulant volume ratio.

Written informed consent was given by all participants. All patients were evaluated and treated in accordance with the latest European Society of Cardiology (ESC) clinical guidelines [24]. The study was conducted in accordance with the Helsinki Declaration. The protocol was approved by the Local Ethics Committee City Clinical Hospital named after V.V. Vinogradov. The identification of patients has been made anonymous for the publication of the article.

### 2.2. Laboratory Assays

Routine coagulation tests. Routine coagulation tests were measured using an ACL TOP 700 coagulometer (Instrumentation Laboratory, Bedford, MA, USA) in platelet-poor plasma (PPP) obtained by whole-blood centrifugation at 1750 g for 15 min. The following assays were performed: *APTT* (SynthASil, Instrumentation Laboratory, Bedford, MA, USA), prothrombin (RecombiPlasTin 2 G, Instrumentation Laboratory, Bedford, MA, USA) and fibrinogen concentration (QFA Thrombin, Instrumentation Laboratory, Bedford, MA, USA).

Thrombin generation assay. The kinetics of thrombin generation in plasma was monitored using the hydrolysis rate of the slow fluorogenic substrate Z-Gly-Gly-Arg-AMC by thrombin formed during clotting, as described in [25]. It was performed in platelet-free plasma (PFP) obtained by serial centrifugation at 1600× *g* for 15 min and at 10,000× *g* for 5 min. Plasma was diluted with a mixture of the fluorogenic substrate, phospholipids and tissue factor diluted in BSA vehicle in a ratio of 2:1.

Thrombodynamics-4d. The thrombodynamics-4d assay was performed with a thrombodynamics-4d analyzer and kit (HemaCore LLC, Moscow, Russia) PFP. This method is based on registering simultaneous fibrin spatial clot growth (via a light-scattering signal) and thrombin spatial distribution (via thrombin-induced hydrolyzation of the fluorogenic substrate) after the activation of clotting in a thin layer of plasma after contact with an immobilized tissue-factor-bearing surface [26]. In this experimental setup, lyophilized reagents were used, and thus, blood plasma was not diluted.

### 2.3. Statistical Analysis

We used the Mann–Whitney U test (non-related samples), which is able to estimate the difference between two independent groups with non-normally distributed data in terms of the level of the parameter, measured quantitatively. It allows for the identification of differences in the value of the parameter between groups of small size. Results were considered significant if *p* < 0.05. Statistical analysis was performed using Origin Pro 2018 (OriginLab Corp., Northampton, MA, USA) software.

### 2.4. Detailed Model of Blood Coagulation

This model describes the biochemical reactions of blood plasma clotting. The chemical design of the model is based on our previous model [27], with modifications. The mathematical model of blood coagulation consisted of 59 volume variables and 19 surface variables for the spatially inhomogeneous case and 78 variables for the homogeneous case. Model parameters, constants and equations are presented in in the Appendix A. An important feature of the developed model of blood coagulation was taking into account the contributions of various types of lipid surfaces to the formation of the system’s response. The model considered 3 types of lipid surfaces: (1) lipids initially present in the blood plasma sample; (2) artificial phospholipids in which the coagulation activator tissue factor is dissolved; and (3) artificial phospholipids added to the sample to enhance the response of the system (thrombin generation test and thrombodynamics-4d).

### 2.5. Pharmacokinetic (PK) Model

The LMWH PK model was based on the assumption that after its subcutaneous introduction, the drug first enters the interstitial space, from where it passes into the bloodstream, where its free form is cleared, while its lipid-bound form is protected from clearance.

LMWH, when administered subcutaneously, enters the interstitial space. There is an exchange with blood, causing a gradual increase in LMWH in the blood. At the same time, it is partially removed from the bloodstream by renal clearance and partially associated with lipids, which leads to a slowdown in its excretion. Model equations and constant values are given in the Appendix A. To take into account the individual characteristics of patients, the following amendments were introduced.

### 2.6. Simulation Output Parameters

During the simulations, the time-dependent spatial distribution of the concentration of each reagent was recorded (distribution of thrombin, fibrin, etc.). The spatiotemporal distribution of fibrin was transformed into the kinetics of the propagation of the clot growth front using the following rules. We assume that clotting occurs as soon as the local fibrin concentration is higher than 50% of the initial fibrinogen concentration [28]. The position of the zone where the fibrin concentration is half the initial value of fibrinogen is the coordinate of the clot growth front. Plotting the movement of the clot growth front coordinate over time, the clot size (CS) as a function of time was obtained. The time point at which CS increased from the zero value was considered the lag time (Tlag). The slope of the CS, linearized in the time interval Tlag + 2 min − Tlag + 6 min, was considered the initial velocity of clot growth (Vi). The slope of the CS, linearized in the time interval Tlag + 15 min − Tlag + 25 min, was considered the stationary velocity of clot growth (Vst). The thrombin spatial distribution is presented as a complex-shaped curve with a prominent peak propagating from the activation site. This peak has an almost constant amplitude (A).

### 2.7. Comsol Parameters of Simulation and Numeric Methods

Simulations were performed with Comsol 6.0 (Comsol, Burlington, MA, USA).

In our 1D simulations, we used an absolute tolerance factor of 10^−6^ (scaled) and a relative tolerance of 0.01. The time-stepping method was generalized alpha (which contains a parameter, called alpha in the literature, to control the degree of damping of high frequencies) with strict steps. Equations were solved using MUltifrontal Massively Parallel sparse direct Solver (MUMPS). We used an area 4 mm long with an equidistant mesh of 400 nodes.

In our homogeneous simulations, we used an absolute tolerance factor of 0.05 (scaled) and a relative tolerance of 0.005.

## 3. Results

### 3.1. Detailed Model of Blood Coagulation: Verification and Evaluation

Native blood contains phospholipid surfaces from the following sources: platelets, chylomicrons, and high-, medium-, low-, and very low density lipoproteins. Since we examined a blood plasma sample obtained from whole blood by double centrifugation, there were practically no platelets in this sample, and their contribution could be neglected. Of the other sources of phospholipid surfaces, procoagulant activity was observed mainly only on very low density lipoproteins [29]. An estimate of the phospholipid surface equivalent for very low density lipoproteins gives a value of the order of 200 μM [30]. This value was used to describe the lipids originally present in the sample.

Tissue factor is a transmembrane glycoprotein and must be incorporated into the phospholipid surface in order to function properly. Therefore, the standard reagent used to activate coagulation in coagulation tests is phospholipid vesicles with incorporated recombinant tissue factor. According to the literature data, in such a reagent, there are about 5000 phospholipid molecules per molecule of tissue factor [31]. These data were used to describe artificial lipids that enter the plasma sample upon the activation of coagulation. Artificial phospholipid vesicles composed of phosphatidylcholine and phosphatidylserine (3:1) are widely used in various clinical and scientific studies of hemostasis to enhance the signal and are a necessary reagent for thrombin generation and thrombodynamics-4d tests. Their standard concentration is 4 µM. The main reactions of blood coagulation occur on phospholipid surfaces, where complexes such as external and internal tenase and prothrombinase are formed. In this case, the proteins forming the complex occupy a certain place on the surface, and if this surface is too small, then the formation of complexes will be difficult. In this work, this was taken into account: on each of the three types of lipids present in the model, the formation of complexes of internal tenase (a complex of factors IXa:VIIIa) and prothrombinase (a complex of factors Xa:Va), as well as the binding of prothrombin to the lipid surface, is possible. On phospholipids containing tissue factor, the formation of an external tenase (complex of factors VIIa:TF) is also possible. The binding of factors to the lipid surface reduces the free (reactive) area, thereby slowing down the binding of other factors to lipids.

The model was verified in a homogeneous formulation using the thrombin generation test (Figure 1). For verification, we compared parameters such as T1/2 (time to reach half of the maximum signal from the fibrin clot), Tmax (time to reach the maximum thrombin concentration) and Amax (maximum thrombin concentration).

The thrombin generation test and thrombodynamics-4d test measure the magnitude of the fluorescent signal from a substrate that is cleaved by thrombin [26]. This substrate can be cleaved not only by free thrombin but also by the thrombin–alpha2-macroglobulin complex. Thus, in the test, we registered a signal from two components. In the model, we can separate the contribution of each component, but in the future, to compare the simulation results and experimental data, we will use the total signal of free and alpha2-macroglobulin-bound thrombin with a factor of 0.6, because bound thrombin is somewhat worse at cleaving the substrate than free thrombin [32].

Figure 1C shows the integral contribution of thrombin and the thrombin–a2-macroglobulin complex, where thrombin actually means the sum of free thrombin and thrombin in complex with a2-macroglobulin. Therefore, panel D also shows the sum of free thrombin and thrombin in complex with a2-macroglobulin.

The simulations differ from the experimental data in the clotting onset and the signal from the thrombin-alpha2–macroglobulin complex. The first strongly depends on the preanalytical conditions [33], which are not described in simulations. The second depends on the endogenous level of alpha2-macroglobulin and may be different in the experimental and simulation setups.

For fibrin generation, in vitro T1/2 = 2.2 ± 0.3 min, and in silico T1/2 = 7.5 min. For thrombin generation, in vitro Tmax = 9.5 ± 0.4 min and Amax = 200 ± 25 nM, and in silico Tmax = 12.2 min and Amax = 132 nM.

In a spatially distributed setting, the model was verified using the thrombodynamics-4d test (Figure 2).

The clot growth lag time was Tlag = 0.87 ± 0.25 min in vitro and Tlag = 0.6 min in silico. The stationary clot growth rate in vitro was V = 34.8 ± 0.53 µm/min, and in silico, the clot growth rate was V = 34.7 µm/min. The amplitude of the thrombin peak was A = 45 nM in vitro and A = 40 nM in silico.

It can be seen that the simulation results in the homogeneous case are in qualitative agreement with the experimental data, while in the spatially inhomogeneous case, a good quantitative agreement is observed (the parameters differ by no more than 15%).

Afterwards, we used experimental data derived from our previous work [34] (Table 1a) to examine the model’s ability to describe individual differences in hemostasis associated with differences in the concentration of coagulation factors using data on thrombin generation and thrombodynamics assays performed in factor-deficient plasmas (Table 1b).

The mean error for the thrombin generation test (for all parameters and deficiencies combined) is 38 ± 50%, and for thrombodynamics-4d, it is 54 ± 55%. As we can see, the simulation and in vitro measured parameters differ by no more than 1.5 times. The discrepancies in experimental and simulation tests may originate from the uncertainty of coagulation factor concentrations in deficient plasma, as only the deficient factor level was estimated. Another source of the difference may be the amounts of preactivated factors (i.e., cold activation of factor VII [35] and contact activation of factor XII [36]) generated in deficient plasmas during production and storage.

### 3.2. PK Model of LMWH: Development and Evaluation

We found that, according to the kinetics of LMWH clearance, patients can be divided into two groups (Figure 3): those whose heparin concentration dropped by more than 60% of the maximum (measured at point 1) and those whose heparin concentration fell by less than 60% of the maximum during the same time.

LMWH clearance in group 1 was significantly higher than in group 2 (Mann–Whitney test, *p* = 0.05). According to the results of the comparison of these patients, it turned out that, in group 1, total cholesterol was significantly lower than in group 2 (Figure 3B) (Mann–Whitney test, *p* = 0.05). The direct comparison of normalized anti-Xa activity and the total cholesterol level revealed that they correlated with r = 0.67 at time point 2 (6 h after LMWH administration) and r = 0.79 at time point 3 (12 h after LMWH administration) (Appendix A).

### 3.3. Pharmacokinetic Model of LMWH

We used individual patients’ parameters (Table 2) to develop a PK model of LMWH clearance.

LMWH is administered subcutaneously, and the drug first enters the interstitial space, from where it passes into the bloodstream. By analogy with models describing the dynamics of glucose changes [37], several compartments were considered in the LMWH model (Figure 4).

LMWH, when administered subcutaneously, enters the Vint compartment. Through exchange with the Vb compartment, its concentration in the blood begins to increase. At the same time, it is partially removed from the bloodstream by renal clearance and becomes partially associated with lipids, which leads to a slowdown in its excretion.

Model equations and constant values are given in the Appendix A. To take into account the individual characteristics of patients, the following amendments were introduced. Vb is the volume of blood flow, calculated based on the sex and values of the weight and hematocrit of the patient according to the formula:
(1)Vb=1.4⋅W⋅0.065⋅(1−HC) for female
(2)Vb=1.4⋅W⋅0.075⋅(1−HC) for male

Vint is the interstitial volume, calculated based on the value of the patient’s weight according to the formula:(3)$$\mathrm{Vint}=\frac{W}{80}\cdot 6$$

Here, *W* is the patient’s weight in kilograms, and *HC* is the hematocrit (%). The rate of association with *Ka* lipids depends on the level of cholesterol:(4)$$Ka=Chol\cdot 0.15\cdot 10^{-5}\frac{1}{s}$$

Here, *Chol* is the measured level of total cholesterol (mmole/L).

The average half-life of LMWH is 4.5–5 h. In the model, the rate of elimination of LMWH depends on the level of creatinine (which describes the patient’s renal clearance status):(5)$$Ke\lim=\left(1+\frac{33}{Creat}\right)\cdot 3.85\cdot 10^{-5}\frac{1}{s}$$

Here, *Creat* is the measured creatinine level (µmole/L).

The model was validated by comparing the simulation results with the measured values of heparin concentration (Figure 5), and the mean error was quite low.

### 3.4. Personalized Hemostasis Profile

The levels of intrinsic pathway coagulation factors were assessed from the results of the *APTT* test. The levels of factors of the external pathway were assessed according to the *PT* test. As shown in [38], the intrinsic clotting factor concentrations (VIII, IX, XI and XII) were inversely proportional to the *APTT* value, while the *PT* value was inversely proportional to the extrinsic and common factor concentrations (V, VII and X) [39].

Thus, we assumed that the concentrations of factors Fg, VIII, IX and XI changed in proportion to the change in *APTT*:(6)$$F=F_{0}\cdot \frac{31.43}{APTT}$$

*F* is the initial factor concentration, *APTT* is the value of the *APTT* test, and 31.43 is the mean value of the normal range of *APTT* values of our test system.

The concentrations of factors II, V, VII and X changed in proportion to the change in *PT*:(7)$$F=F_{0}\cdot \frac{12.09}{PT}$$

*F* is the initial factor concentration, *PT* is the value of the *PT* test, and 12.09 is the mean value of the normal range of *PT* values of our test system.

*F*_0_ is the normal average concentration of factor *F*.

The antithrombin concentration was measured directly.

The value of very low density lipoproteins changed in proportion to the change in the value of LDL. The concentration of LMWH in the sample was calculated using the pharmacokinetic model.

Evaluation: Experimental data vs. simulation.

Figure 6 shows the comparison of the distribution profiles of thrombin and the dependence of clot size on time in the experiment and in the simulation for patient 2 at point 1, corresponding to 3 h after LMWH administration. Table 3a–d show the comparison of the parameters of the thrombodynamics-4d test and the simulation results for patient 2 at time points 1, 2 and 3, corresponding to 3 h, 6 h and 12 h after LMWH administration.

The main difference between the in vitro experiment and the simulation is that clot propagation is almost arrested after one hour in the patient’s plasma, while in silico, it continues without a velocity drop. As the propagation rate depends on the levels of factors VIII, IX and, to a lesser extent, XI, it may indicate that our method for their concentration estimation may not be very accurate, and further improvement is needed. However, as most of the test parameters were evaluated before 60 min of clotting, the discrepancy in the simulated and measured parameters is reasonably small.

## 4. Discussion

One of the variants of using computer modeling in diagnostics was demonstrated in the works of K.E. Brummel-Ziedins’ group. They suggested that combinations of variations in the concentrations of clotting factors and inhibitors can lead to abnormal clotting, even if all concentrations are in a normal range separately. They measured the levels of coagulation factors, simulated the generation of thrombin in the model, and compared it with the experimental data [14,40,41,42]. This approach demonstrated that such combinations can indeed explain bleeding and prothrombotic phenotypes.

In a study by another group [43] using the model in [13], the authors analyzed the quantitative effects of blood plasma dilution on the generation of thrombin in the context of intersubject variability. Using data from the LETS study [44], in which the concentrations of clotting factors were measured in 472 healthy volunteers, the authors simulated thrombin generation in undiluted, 2-, 3- and 5-fold-diluted plasma. Dilution caused a decrease in the thrombin peak height, the area under the curve and the maximum rate of thrombin generation. In another study [18] by this group, the effect of a reduced temperature on thrombin generation was investigated. For this purpose, the authors created a set of random temperature coefficients for each of the model parameters and compared the calculated thrombin generation curves with the range in which they were supposed to lie. The authors showed that hypothermia in the range from 31 to 36 degrees Celsius slowed the generation of thrombin, reduced the maximum rate of thrombin generation and had almost no effect on the thrombin peak height or the area under the curve.

Existing studies use the thrombin generation test to assess the state of the blood coagulation system, which does not take into account the spatial distribution of clotting reactions or the transport of coagulation factors.

In this work, we propose a systems biology approach for personalized hemostasis correction. This approach is based on the combination of a detailed biochemical model of blood coagulation, which was validated to describe thrombin and fibrin formation in global hemostasis assays, such as the thrombin generation test or thrombodynamics, under normal conditions or in the case of coagulation factor deficiency. Coupled with information on the patient’s current clotting factor concentrations, estimated from the results of the routine clotting tests *APTT* and *PT*, we were able to describe the output of the global tests, thus indirectly estimating the current patient’s hemostasis.

The detailed model of blood coagulation employed was verified to describe blood coagulation in a wide range of initial conditions, such as normal concentrations of coagulation factors and their deficiency. In addition, this model included test-specific parameters, such as the dilution of the plasma sample, amounts of artificial lipids, fluorogenic substrate and tissue factor reagent concentration. An ordinary differential equation model was used to describe the thrombin generation assay, where all reactions occurred in the mixed volume, while a partial differential equation model with a diffusion component was used to describe the thrombodynamics-4d assay, where the tissue-factor-bearing surface activated clotting. The model employed three types of phospholipid surfaces that can be present in the blood coagulation assay: very low density lipoproteins (patient-dependent); artificial phospholipids, which can be added to the plasma sample in the thrombin generation and thrombodynamics-4d assays (assay-dependent); and artificial lipids, which are a component of the tissue factor reagent (assay-dependent). However, this model, as well as an in vitro global test that utilizes blood plasma samples to estimate the coagulation state, lacks information about the condition of blood cells, endothelium and vessel patency and is able to consider only disorders of blood plasma clotting, as well as their correction, but in the other cases, different models and tests should be used.

The global hemostasis thrombodynamics assay employs PFP that is not supplemented with exogenous phospholipids to simulate spatial clot growth [45]. In this test, TF is bound to the surface, and clotting in the bulk of plasma occurs only due to procoagulant microparticles present in the sample. The procoagulant microparticle count in PFP can be as high as 3500–10,000/µL and between 0.35 and 1.0 µm in diameter [46]. Considering their average diameter of 0.8 µm and taking into account that one phospholipid head group is 0.7 nm^2^ [47], we can calculate that the phospholipid concentration in PFP is about 4–12 nM. In both of our global hemostasis assays, the thrombin generation test and thrombodynamics-4d, the plasma sample is supplemented with 2–4 µM of artificial phospholipids, which is almost 3 orders of magnitude higher than the phospholipid amount from microparticles. The amount of lipids in the TF reagent is about 8000 higher than TF [48,49], which corresponds to a 40 nM phospholipid concentration in the case of the standard 5 pM TF dose in the thrombin generation test in the case in which no artificial lipids are added. Due to this, we neglected the impact of microparticles on blood coagulation in our simulations.

The patient-dependent coagulation factor level was estimated based on the results of routine hemostasis tests (*APTT* and *PT*). This approach let us imitate the results of the global hemostasis assays (thrombin generation and thrombodynamics-4d) for each patient with relatively good accuracy.

Analyzing anti-Xa activity in the group of patients, we found that LMWH clearance was significantly lower in the group with high total cholesterol levels. In order to describe the efficacy of LMWH therapy, we developed a personalized pharmacokinetic model of LMWH clearance. Existing pharmacokinetic models of LMWH are population-based (i.e., do not take into account the individual characteristics of patients) [50]. According to the literature, the main route of elimination of LMWH from the body is renal clearance [51]. However, there is evidence that heparin can bind to apoproteins on very low density lipids [52] and apoprotein E [53]. When developing the pharmacokinetic model of LMWH, we proceeded from the assumption that the clearance of LMWH depends on the level of cholesterol, but this assumption needs additional testing in a larger group of patients in a separate study. Our model described the observed level of LMWH in patients from both groups, with high or low total cholesterol levels, quite well. Embedding this PK model in our detailed biochemical model of blood coagulation let us describe the current patient’s state under LMWH therapy. However, the limitations of our model should be mentioned. Our estimations of blood and interstitial volumes are not very accurate and do not take into account extreme values of body weight, the body’s tissue composition or impaired kidney function.

We found that the highest deviation of the simulation parameters from the in vitro measured parameters was for Tlag and A (value of the peak of thrombin propagating impulse). However, for each patient, the mean deviation, combined for all parameters, was relatively low.

## 5. Conclusions

In silico simulations instead of real hemostasis tests may reduce the financial and time costs of treatment and facilitate therapy adjustments, as it may predict the hemostasis state based on the individual parameters of the patient. Our approach, described for LMWH, may be used for other hemostasis-correcting drugs, such as anti-hemophilia drugs, oral anticoagulants, etc. A possible further step can be the combination of the simulation of plasma hemostasis with more sophisticated models of platelet hemostasis, which may improve the prediction accuracy and widen the outcome parameter spectrum.

## Figures and Tables

**Figure 1 jpm-12-01903-f001:**
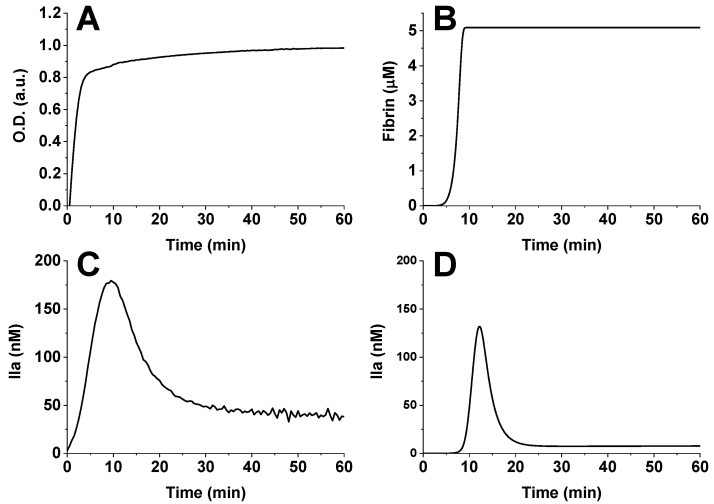
Verification of the blood coagulation model in a homogeneous formulation. (**A**) In vitro fibrin generation, normal pooled plasma from healthy donors. Optical density at wavelength 605 nm was measured. (**B**) In silico fibrin generation. (**C**) In vitro thrombin generation, normal pooled plasma from healthy donors. (**D**) In silico thrombin generation. Coagulation was activated by 5 pM tissue factor in the presence of 2 μM phospholipid vesicles and 400 μM fluorescent substrate of thrombin.

**Figure 2 jpm-12-01903-f002:**
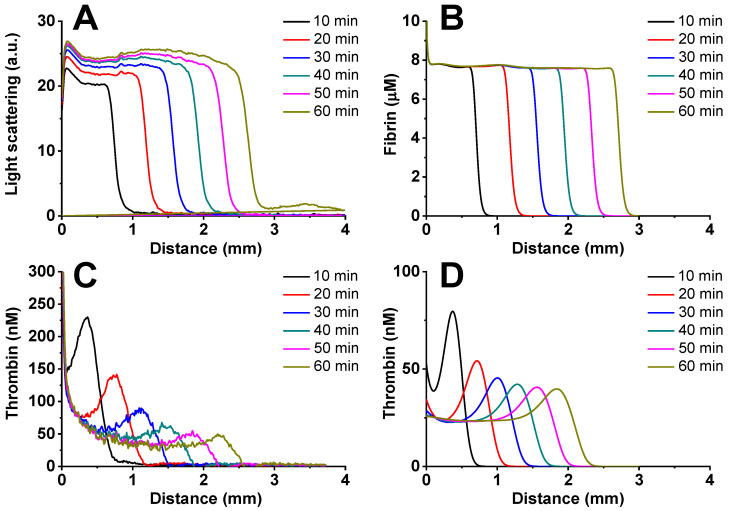
Verification of the blood coagulation model in a spatially distributed formulation. (**A**) In vitro fibrin generation, normal pooled plasma from healthy donors. Light-scattering signal from 625 nm LED light source was measured. (**B**) In silico fibrin generation. (**C**) In vitro thrombin generation, normal pooled plasma from healthy donors. (**D**) In silico thrombin generation. Coagulation was activated by 100 pmol/m^2^ tissue factor in the presence of 2 μM phospholipid vesicles and 400 μM fluorescent substrate of thrombin. A typical experiment is shown.

**Figure 3 jpm-12-01903-f003:**
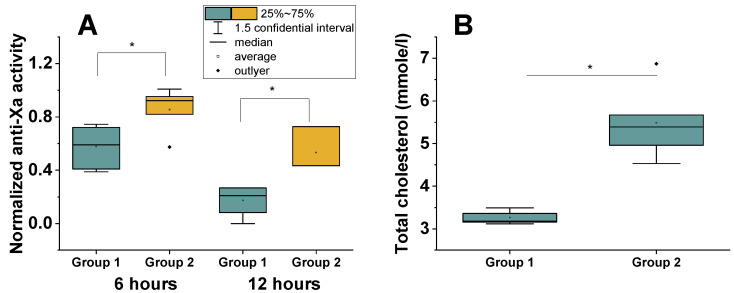
Time course of anti-Xa activity depends on total cholesterol level. (**A**) Normalized anti-Xa activity. The measured anti-Xa activity normalized to the level of antithrombin. The level at point 1 (3 h after administration) was taken as 100% for each patient. Group 1 and group 2 differ, with a significance of 0.05 (*) (Mann–Whitney test). (**B**) Total cholesterol in group 1 was significantly lower than in group 2 (Mann–Whitney test, *p* = 0.05 (*)).

**Figure 4 jpm-12-01903-f004:**
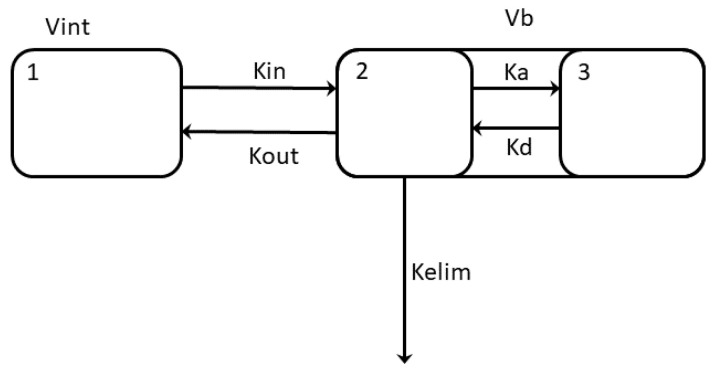
Scheme of the pharmacokinetic model of LMWH. Vint—interstitial volume. Vb is the volume of blood flow. An exchange takes place between Vint and Vb compartments, and LMWH passes from one to the other at rates described by the constants Kin and Kout. Within the Vb compartment, LMWH can be in one of two states: free (2) and associated with lipids (3). The transition from one state to another is described by the constants Ka and Kd. Free LMWH is removed from the bloodstream at a rate described by the Kelim constant.

**Figure 5 jpm-12-01903-f005:**
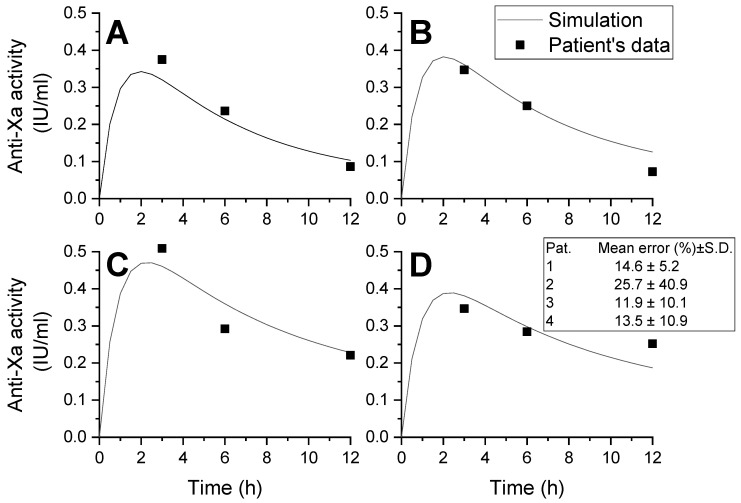
Comparison of simulation results of LMWH pharmacokinetics and measured data. (**A**) Patient 1; (**B**) patient 2; (**C**) patient 3; (**D**) patient 4. Patient 1’s total cholesterol level was 3.36 mmol/L and patient 2’s total cholesterol level was 3.12 mmol/L, both of whom were from the low-cholesterol group; patient 3’s total cholesterol level was 5.67 mmol/L, and patient 4’s total cholesterol level was 4.96 mmol/L, both of whom were from the high-cholesterol group. The inset shows the mean error of simulation for each patient, and it was mostly within 15%.

**Figure 6 jpm-12-01903-f006:**
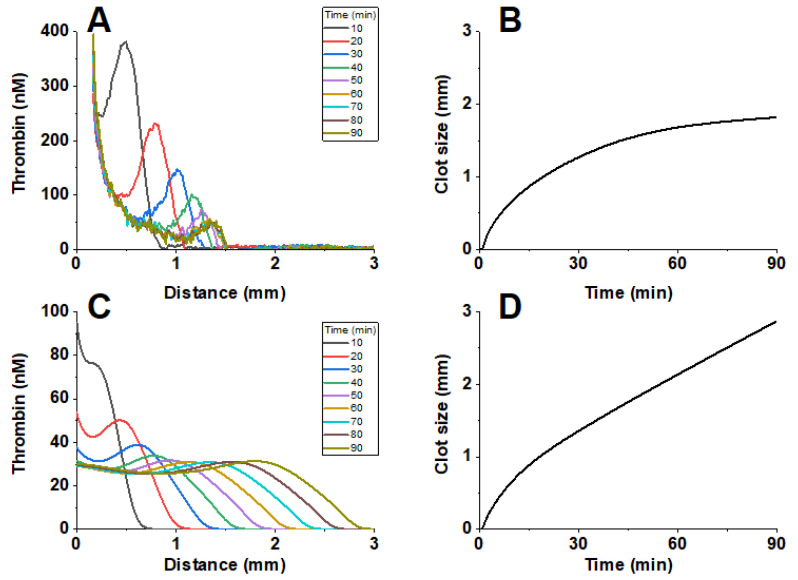
Comparison of thrombodynamics-4d test and simulation results for patient 2, sample 1 (3 h after LMWH administration). (**A**) Thrombin profiles in the experiment. (**B**) The clot size versus time in the experiment. (**C**) Profiles of free thrombin and thrombin associated with alpha2-macroglobulin in the simulation. (**D**) The clot size versus time in the simulation.

**Table 1 jpm-12-01903-t001:** (a) Parameters of in vitro thrombin generation and thrombodynamics-4d assays. (b) Parameters of in silico thrombin generation and thrombodynamics-4d assays.

**(a)**					
**Deficient Factor**	**Thrombin Generation Assay**		**Thrombodynamics-4d**		
	**T** _max_	**A** _max_	**T** _lag_	**V**	**A**
	**min**	**nM**	**min**	**µ/min**	**nM**
FII (<2%)	12	15	3.2	29.1	7
FV (<1%)	NC	NC	14.2	20.1	NP
FVII (<1%)	NC	NC	17.3	26.8	24
FVIII (<1%)	18	45	0.6	18.0	NP
FIX (<1%)	10	94	0.9	18.2	NP
FX (<1%)	NC	NC	49.3	8.4	NP
FXI (<1%)	10	62	0.6	36.0	25
**(b)**					
**Deficient Factor**	**Thrombin Generation Assay**		**Thrombodynamics-4d**		
	**T_max_**	**A_max_**	**T_lag_**	**V**	**A_peak_**
	**min**	**nM**	**min**	**µ/min**	**nM**
FII (2%)	12	3	1.4	34	1
FV (<1%)	NC	NC	3.2	27.6	0.4
FVII (<1%)	NC	NC	5.9	42	55
FVIII (<1%)	30	25	0.6	7.2	NP
FIX (<1%)	24	46	0.6	8.4	10
FX (<1%)	NC	NC	10.3	3.6	NP
FXI (<1%)	11.5	143	0.6	34.5	25

NC—no clotting; NP—no thrombin peak.

**Table 2 jpm-12-01903-t002:** Patients’ characteristics.

No.	Sex	Age (y)	Weight (kg)	LMWH Drug	Diagnosis	Creatinine (µmol/L)	Total Chole-sterol (mmol/L)	High-Density Lipoproteins (mmol/L)	Low-Density Lipoproteins (mmol/L)	Hematocrit (%)
1	M	72	80	Clexane	CFH	32	3.36	1.08	1.32	38.8
2	M	74	92	Clexane	COPD	38	3.12	0.75	1.94	47.7
3	M	57	96	Clexane	CHD	79	5.67	0.87	2.78	56.3
4	M	67	77	Clexane	CHD	90	4.96	0.82	3.21	38
5	F	18	53	Clexane	D t1	89				32.4
6	F	87	78	Arixtra	CHD	47	4.53	0.91	2.93	47.5
7	M	79	82	Clexane	CHD	83	3.49	0.8	2.42	23.1
8	M	62	92	Clexane	ALE	83	3.18	0.65	2.01	19.8
9	M	64	79	Clexane	CHD	40	3.16	0.75	1.9	51
10	F	18	46	Clexane	D t1	111				31.3
11	M	75	86	Clexane	CHD	51	5.39	1.42	2.6	51.1
12	M	59	89	Clexane	CHD, VT	55	6.87	0.94	4.62	40.7

CFH—Chronic Heart Failure; COPD—Chronic Obstructive Pulmonary Disease; CHD—Coronary Heart Disease; D t1—diabetes type 1; ALE—Atherosclerosis of Lower Extremities; VT—Ventricle Tachycardia.

**Table 3 jpm-12-01903-t003:** (a). Comparison of in vitro parameter Tlag of thrombodynamics-4d and results of computer simulations. (b). Comparison of in vitro parameter Vi of thrombodynamics-4d and results of computer simulations. (c). Comparison of in vitro parameter Vst of thrombodynamics-4d and results of computer simulations. (d). Comparison of in vitro parameter A of thrombodynamics-4d and results of computer simulations.

**(a)**				
**Patient**	**Time Point**	**In Vitro Tlag (min)**	**In Silico Tlag (min)**	**Deviation (%)**
2	1	1.2	0.7	42
2	2	1.1	0.8	27.3
2	3	1.5	0.8	46.7
3	1	0.9	0.9	0
3	2	1.1	0.8	27.3
3	3	1	0.8	20
1	1	0.9	0.7	22.2
1	2	0.9	0.7	22.2
**(b)**				
**Patient**	**Time Point**	**In Vitro Vi (µm/min)**	**In Silico Vi (µm/min)**	**Deviation (%)**
2	1	64.5	69.6	7.9
2	2	64.4	72.6	12.7
2	3	67.2	77.4	15.2
3	1	64.9	63	2.9
3	2	66.4	67.2	1.2
3	3	68.8	71.4	3.8
1	1	71.2	71.4	0.3
1	2	81.6	75.6	7.3
**(c)**				
**Patient**	**Time Point**	**In Vitro Vst (µm/min)**	**In Silico Vst (µm/min)**	**Deviation (%)**
2	1	27.6	31.9	15.6
2	2	28.7	34.9	21.6
2	3	35.6	39.2	10.1
3	1	27.3	27	1.1
3	2	29.9	29.6	1
3	3	32.3	33.4	3.4
1	1	30.7	34.8	13.3
1	2	42.6	38.3	10.1
**(d)**				
**Patient**	**Time Point**	**In Vitro A (nM)**	**In Silico A (nM)**	**Deviation (%)**
2	1	52.2	31	40.6
2	2	50.4	42	16.7
2	3	134	55	58.9
3	1	33.7	15.4	54.3
3	2	25.1	22.7	9.6
3	3	23.9	32.5	36
1	1	48.3	44	8.9
1	2	-	55	-

Mean deviation (±SD) was 26 ± 14.3% for Tlag, 6.4 ± 5.4% for Vi, 9.5 ± 7.4% for Vst and 32.1 ± 20.7% for A. For patient 2, mean deviation (±SD) (for all parameters combined) was 26.3 ± 16.7%. For patient 3, mean deviation (±SD) (for all parameters combined) was 13.4 ± 17.5%. Thus, we assume that our simulations are consistent with the in vitro data.

## Data Availability

Data are available on request from the authors.

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
