# Peer review of "Systems Biology Approach for Personalized Hemostasis Correction"

_jpm, 2022, doi:10.3390/jpm12111903_

Round 1

Reviewer 1 Report

In this paper, the authors proposed a new model to define the hemostasis profile and/or to evaluate the efficacy of LMWH treatment of patients. 

The article would be of interest, particularly because this approach would save time and money. In my opinion, another strength that should be underlined more is the high reproducibility of an in silico approach.

My major concern is the poor sample size. The study would increase its impact if the approach were applied to a greater number of samples. For the same reason, the number of healthy controls must be increased, at least to equal the number of patients.  

Anyway, the study sounds appealing but minor revisions are required. 

1) to give strength to the results, tables containing single patient analysis should be moved to the main text. 

2) In some cases, the "results" section is too schematic, resulting in a description of the Figure more than the description of the analytic process leading to the result. (Paragraph 3.4 is the best example, particularly in its final part). I suggest adding a small sentence the introduce the aim of the "experiment"

3) The introduction is too long. I would suggest the authors summarise it.

Author Response

In this paper, the authors proposed a new model to define the hemostasis profile and/or to evaluate the efficacy of LMWH treatment of patients. 

The article would be of interest, particularly because this approach would save time and money. In my opinion, another strength that should be underlined more is the high reproducibility of an in silico approach.

My major concern is the poor sample size. The study would increase its impact if the approach were applied to a greater number of samples. For the same reason, the number of healthy controls must be increased, at least to equal the number of patients.

Thank you for the detailed evaluation of our work. Indeed, the size of our experimental group is relatively small, which of course decrease the significance of our results regarding the connection of the total cholesterol level and LMWH clearance. Our further investigation will incorporate a larger set of patients undergoing LMWH therapy.

Anyway, the study sounds appealing but minor revisions are required. 

1) to give strength to the results, tables containing single patient analysis should be moved to the main text. 

2) In some cases, the "results" section is too schematic, resulting in a description of the Figure more than the description of the analytic process leading to the result. (Paragraph 3.4 is the best example, particularly in its final part). I suggest adding a small sentence the introduce the aim of the "experiment"

3) The introduction is too long. I would suggest the authors summarise it.

According to your concerns, we have moved the table with patients’ data to the main text, expanded description of the results, and shortened the introduction.

Reviewer 2 Report

In this manuscript the concept of using a system-biology approach to predict hemostasis and its response to therapeutic interventions are interesting and potentially-cost saving. Nevertheless, the findings are too preliminary to support the conclusion that the simulation “can be used for assessment of the dosage and/or administration of other hemostatic correction drugs.”

Figure 1 and Table 1a/1b: The data as presented in Figure 1 and Tables 1a/1b shows large discrepancies that raise serious concerns of reliability.  

Figure 2: The reasons for representing fibrin generation in different units in panels A and B should be clarified. In consistence with what has been reported, the data of this study suggests thrombodynamics-4d is more reliable than thrombin generation test.

Figure 3: The correlation data between LMW heparin and cholesterol levels is novel and interesting but quite preliminary. Furthermore, without a formal regression analysis, it is difficult to interpret the data. A P value of 0.05, based on a small data set and without a regression analysis, should not be stretched to the claim that the “total cholesterol in group 1 was significantly lower than in group 2” (Figure 3 legend).

Figures 4 and 5: The limitations of the pharmacokinetic model as presented in Figure 4 should be mentioned. Firstly, the determination of blood volume based on body weight is problematic, as muscles and adipose tissues can differ vastly in vascularity.  Neither body weight nor BMI addresses this problem. Secondly, some patients may have acute kidney injury or are recovering from it. Thirdly, the interstitial volume is not only determined by body weight but also by variations in the intravascular oncotic pressure and vascular permeability, both of which, for various reasons, may shift during the course of anticoagulation therapy. Extreme body weight, unstable or impaired kidney function, and shifting interstitial edema are the conditions for which anti-Xa measurements are most commonly ordered in clinical practice. Unfortunately, the limitations of the simulation indicate that the model is unlikely to reliably replace direct anti-Xa measurement for patients with any of these conditions.  

In fact, as has been reported, it is doubtful that any of the global tests of hemostasis are better than direct measurement of the target therapeutic action (doi: 10.1371/journal.pone.0199900). Similarly, the authors should clarify if there are any global tests that have been demonstrated to provide a better prediction of DVT in patients with deficiency of a coagulation factor inhibitor or bleeding in patients with a coagulation factor deficiency.

It should also be noted that, none of the in-vitro global tests or simulations addresses the critical roles of circulatory stasis, overt or sub-clinical vessel injury and perturbation of vascular endothelial cells and blood cells in the development of thrombosis and/or hemorrhages. Attention these non-fluid factors are as essential if not more so for preventing thrombosis or hemorrhage in patients with or without a hypercoagulability or bleeding diathesis. 

Author Response

In this manuscript the concept of using a system-biology approach to predict hemostasis and its response to therapeutic interventions are interesting and potentially-cost saving. Nevertheless, the findings are too preliminary to support the conclusion that the simulation “can be used for assessment of the dosage and/or administration of other hemostatic correction drugs.”

Thank you for the thorough review of our work. Indeed, the findings of our work are too preliminary, and can’t be used for assessment of the dosage of hemostasis correction drug just as is. We have corrected the phrasing in the to underline the possibility of such approach, after better evaluation, to achieve this goal. Now, it is said: “which potentially can help with hemostasis corrections based on the output of the global tests”.

Figure 1 and Table 1a/1b: The data as presented in Figure 1 and Tables 1a/1b shows large discrepancies that raise serious concerns of reliability.

We have added the explanation of the possible origins of the discrepancies between the simulations and in vitro data, presented in the Fig.1 and Tables 1a&b.

After Figure 1, we have added this text: “The simulations differ from the experimental data in the clotting onset and the signal form the thrombin-alpha2-macroglobulin complex. The first strongly depends on the preanalytical conditions (10.1111/jth.12012), which are not described in simulations. The second depends on the endogenous level of alpha2-macroglobulin and may be different in the experimental and simulation setups”.

After tables 1a&b we have added this text: “The discrepancies in experimental and simulation tests may originate from uncertainty of coagulation factors concentration in deficient plasma, as only deficient factor level was estimated. Another source of the difference may be the amounts of preactivated factors (i.e. cold activation of factor VII (10.1055/s-0038-1647755), contact activation of factor XII (10.1055/s-0036-1598003)), generated in deficient plasmas during production and storage”.

Figure 2: The reasons for representing fibrin generation in different units in panels A and B should be clarified. In consistence with what has been reported, the data of this study suggests thrombodynamics-4d is more reliable than thrombin generation test.

In fig.2 we used light scattering signal to measure fibrin in vitro, that is why we used arbitrary units for its concentration on the panel A. This signal is linearly proportional to fibrin concentration (10.1111/j.1538-7836.2005.01128.x). On the panel B fibrin concentration was calculated in silico, and we used mol/L units. We have added this information in legend of fig.2.

Figure 3: The correlation data between LMW heparin and cholesterol levels is novel and interesting but quite preliminary. Furthermore, without a formal regression analysis, it is difficult to interpret the data. A P value of 0.05, based on a small data set and without a regression analysis, should not be stretched to the claim that the “total cholesterol in group 1 was significantly lower than in group 2” (Figure 3 legend).

We performed linear regression analysis to figure out dependence of anti-Xa activity on the total cholesterol level at time points 2 and 3 (6 and 12 hours after LMWH administration) and calculated their correlation coefficients, which were 0.67 and 0.79, respectively. Thus, we could conclude that anti-Xa activity depended on the total cholesterol level, and in the group of patients with higher cholesterol it was elevated, in other words, LMWH clearance was diminished in this group. We have added this information after figure 3, and put the regression plots in the supplement (fig.s1).

Figures 4 and 5: The limitations of the pharmacokinetic model as presented in Figure 4 should be mentioned. Firstly, the determination of blood volume based on body weight is problematic, as muscles and adipose tissues can differ vastly in vascularity.  Neither body weight nor BMI addresses this problem. Secondly, some patients may have acute kidney injury or are recovering from it. Thirdly, the interstitial volume is not only determined by body weight but also by variations in the intravascular oncotic pressure and vascular permeability, both of which, for various reasons, may shift during the course of anticoagulation therapy. Extreme body weight, unstable or impaired kidney function, and shifting interstitial edema are the conditions for which anti-Xa measurements are most commonly ordered in clinical practice. Unfortunately, the limitations of the simulation indicate that the model is unlikely to reliably replace direct anti-Xa measurement for patients with any of these conditions.

We have added the information about limitations of our PK model of LMWH clearance, as well as the limitations of the overall concept of utilization of global hemostasis assays based on plasma sampling in the Discussion.

“Yet, this model, as well as in vitro global test that utilize blood plasma samples to estimate the coagulation state, lacks information about condition of blood cells, endothelium, vessel patency, and is able to consider only disorders of blood plasma clotting, as well as their correction, but in the other cases different models and tests should be used”.

“The limitations of our model should be mentioned. Our estimation of blood and interstitial volumes are not very accurate and does not take into account the extreme values of body weight, or the body’s tissues composition; or impaired kidney functioning”.

In fact, as has been reported, it is doubtful that any of the global tests of hemostasis are better than direct measurement of the target therapeutic action (doi: 10.1371/journal.pone.0199900). Similarly, the authors should clarify if there are any global tests that have been demonstrated to provide a better prediction of DVT in patients with deficiency of a coagulation factor inhibitor or bleeding in patients with a coagulation factor deficiency.

It should also be noted that, none of the in-vitro global tests or simulations addresses the critical roles of circulatory stasis, overt or sub-clinical vessel injury and perturbation of vascular endothelial cells and blood cells in the development of thrombosis and/or hemorrhages. Attention these non-fluid factors are as essential if not more so for preventing thrombosis or hemorrhage in patients with or without a hypercoagulability or bleeding diathesis.

Thrombodynamics can detect LMWH activity as good as its direct measurement, but besides it thrombodynamics can supply the investigator with much more information about overall hemostasis state of the sample, and in the case of coagulation disorder it may help to locate the place of malfunction, i.e. in the reactions, responsible for clotting initiation, or propagation. Use of thrombodynamics helped to improve prediction of VTE in postoperative patients based on the Carpini score (https://doi.org/10.1016/j.jvsv.2019.06.015).

Reviewer 3 Report

The article presents the comparison of measured parameters of blood coagulation with the simulation results of the thrombodynamics-4d assays in the patients undergoing LMHW therapy. The authors also developed the pharmacokinetic model of LMHW and compared them with the results of Anti-Xa activity measurements.

After reading the abstract, I thought the manuscript would be more revealing and innovative. Despite the fact that a large number of results have been presented, they are not very convincing. This is mainly due to the relatively small size of the study group (n = 12 for patients and n = 6 for the control group) and the lack of consideration of the role of platelets in the coagulation model. I appreciate the didactic aspect, where the authors try to explain the issues in detail, but it gives the impression of a redundant and lengthy description - especially for the experts familiar with the blood clotting. In my opinion, the presented manuscript, after the reduction, would be more newsworthy as a preliminary study or a brief report. It is also worth considering to increase the number of measurements of anti-Xa activity and total cholesterol - increase the group of patients, and establish the correlation between the obtained results.

Overall, the study approach is interesting and the methods used are appropriate for most parts, but some observations need to be addressed:

- the results section is redundant (especially the verification the model of blood coagulation, some aspects rather are connected with Methods)

- limitations in the application of the model should be discussed, especially as the Authors ignore the role of platelets (According to recommendation patients who are to receive any heparin should have a baseline platelet count)

- panel E in Figure 6 is not presented

- the statistical analysis is poorly described. It is written so that reader has to guess what the authors meant, i.e: "Mann-Whitney U-test (non related samples)" - .... was used to compare differences between two independent groups when the data were not normally distributed.

Author Response

The article presents the comparison of measured parameters of blood coagulation with the simulation results of the thrombodynamics-4d assays in the patients undergoing LMHW therapy. The authors also developed the pharmacokinetic model of LMHW and compared them with the results of Anti-Xa activity measurements.

After reading the abstract, I thought the manuscript would be more revealing and innovative. Despite the fact that a large number of results have been presented, they are not very convincing. This is mainly due to the relatively small size of the study group (n = 12 for patients and n = 6 for the control group) and the lack of consideration of the role of platelets in the coagulation model. I appreciate the didactic aspect, where the authors try to explain the issues in detail, but it gives the impression of a redundant and lengthy description - especially for the experts familiar with the blood clotting. In my opinion, the presented manuscript, after the reduction, would be more newsworthy as a preliminary study or a brief report. It is also worth considering to increase the number of measurements of anti-Xa activity and total cholesterol - increase the group of patients, and establish the correlation between the obtained results.

Overall, the study approach is interesting and the methods used are appropriate for most parts, but some observations need to be addressed:

- the results section is redundant (especially the verification the model of blood coagulation, some aspects rather are connected with Methods)

Thank you for pointing this out, but we do assume that in order to simulate individual patient’s hemostasis, we need to demonstrate that the model we use can describe the overall behaviour of the coagulation system under normal conditions, and its verification is mostly the only way to do that. As this model is our tool, its proper description is a part of the work, thus we suppose that its verification should be included in the results section.

- limitations in the application of the model should be discussed, especially as the Authors ignore the role of platelets (According to recommendation patients who are to receive any heparin should have a baseline platelet count)

Thank you, we have added the information about the limitations of the model in the Discussion.

- panel E in Figure 6 is not presented

Thank you, we have corrected it.

- the statistical analysis is poorly described. It is written so that reader has to guess what the authors meant, i.e: "Mann-Whitney U-test (non related samples)" - .... was used to compare differences between two independent groups when the data were not normally distributed.

Thank you, we have added the information why we used Mann-Whitney U-test in the main text (because it allows to compare groups of small size).